# ESD Ideas: Near real-time preliminary detection of carbon dioxide source and sink areas using a Laplacian filter

Yana Savytska[1], Viktor Smolii[2], Nils Weitzel[3]

[1]University of Tübingen, Germany, [2]National University of Life and Environmental Sciences of Ukraine, Ukraine, [3]University of Bristol, United Kingdom of Great Britain

*Correspondence to*: Yana Savytska (savytska.yana@mnf.uni-tuebingen.de)

**Abstract.** The constant rise in atmospheric $CO_2$ concentrations is warming the planet and causing climate change. Here, we propose digital filtration with a Laplacian filter for preliminary detection of areas with different changes in the characteristics of natural processes, using $CO_2$ sinks and sources as an example. This approach may improve $CO_2$ monitoring capabilities and enable near real-time detection of $CO_2$ sources and sinks.

Over the past few decades, anthropogenic greenhouse gas (GHG) emissions have led to clearly detectable surface warming (IPCC, 2023). The major part - 75% of all GHGs (Xiao et al., 2016) - is atmospheric carbon dioxide ($CO_2$). Our primary research, therefore, focuses on the development of a new method for $CO_2$ reduction. As part of this method, we propose an algorithm for the near real-time preliminary detection of $CO_2$ source and sink areas. This algorithm can help to facilitate the monitoring, reporting and verification of $CO_2$ source and sink areas. This includes the identification of an area as a $CO_2$ source or sink and it's localization. We test the proposed algorithm using two types of $CO_2$ data measured at the near-surface layer. We applied digital filtration (Burger and Burge, 2016) to a $CO_2$ concentration (CDC) dataset to detect sink and source areas and $CO_2$ flux data to verify the results. Identifying the type of area as a $CO_2$ sink or source could help to improve the usability and functionality of $CO_2$ monitoring services, e.g. the Copernicus Atmosphere Monitoring Service, the NASA Carbon Monitoring System, or to assess the role and efficiency of different ecosystems in the global carbon cycle.

Applying digital filtration to $CO_2$ sinks and sources preliminary detection can be challenging due to their nature and behaviour. Industrial objects have more stable emission characteristics. Natural objects have a clear seasonal and also daily periodic dependence. This leads to the need for continuous observations in near real-time mode. Another potential challenge for satellite datasets are technical limitations in the resolution of satellite datasets (the resolution of a sensor), which indirectly challenge the preliminary detection. At the current stage of our work, we do not focus on the reasons that may affect the accuracy of detection, but aim to explore the ability of digital filters to capture and detect changes in various characteristics of natural processes, for example, for the preliminary detection of $CO_2$ sinks and sources.

The response of an ecosystem to external and internal disturbances is reflected in the carbon balance (CB) of its sources and sinks (Xiao et al., 2016). Recent studies have described ecosystem responses to disturbances using functional indices – NDVI (Liu et al., 2022), NPP, GPP (Mahecha et al., 2022), SIF (Li et al., 2022), biodiversity (Mahecha et al., 2022) and others in the

complex multivariate models (Holm et al., 2023). This makes them potentially accurate but also more resource intensive, less straightforward and less sensitive to short-term changes. Therefore, we propose the CDC as an integral parameter for the near real-time detection of $CO_2$ sources and sinks that can also be applied to long-term observations.

Existing $CO_2$ monitoring services provide spatially distributed CDC on a global scale (Weir and Ott, 2022; CAMS, 2020). This does not include the detection of local $CO_2$ sink and source areas. A possible solution could be an edge detection using digital filtration. This could sharpen the boundaries and make it possible to detect the $CO_2$ sink and source areas with a size corresponding to the resolution of the CDC dataset. Digital filtration is a well-known tool also used in Geosciences, for example, to detect plumes of burning biomass (Goudar et al., 2023). In our paper, we do not quantify $CO_2$ sources and sinks

because quantification is valuable for understanding the consequences of $CO_2$ changes after these changes have occurred. Our focus is on short-term (e.g., hours) $CO_2$ changes, which can help detect $CO_2$ sources and sinks and their different phases of development in near real-time, until further analyses can be performed.

Before applying digital filtration, we need to consider the size of the areas, the characteristics of the internal physical, chemical and biological processes, and the CB of each area. We work with the concept of a "small area" as a cell whose size depends

on the inertia rate of chemical and physical processes, and interpret it as a closed ecosystem based on the characteristics described below. Here the term "small area" is an analogue of "small ecosystem", defined by a set of characteristics and their values that describe an ecosystem in all its parts with a slight (or within the specified range) deviation. This deviation can be neglected at any time and any place within the ecosystem.

The carbon balance can be seen as a strictly hierarchical system in which lower level subsystems separately describe the CB

in terms of its environmental and other conditions. The components of the subsystems are spatially distributed, defining the unique set of components of each area and determining the variability of environmental characteristics in different areas. To identify fluxes in the upper atmospheric CB, we use two principles. The first is the direction of $CO_2$ flows (suffixes "In" and "Src" into the atmosphere or "Out" and "Sink" - out of it). The second principle is relative to the boundary of the area - the prefix "Env" for the external environment and "Int" for internal processes and objects. Accordingly, we describe the total CB

of the area of interest by Eq. (1):

$$CB = EnvIn - EnvOut + \sum IntSrc_k - \sum IntSink_l \qquad (1)$$

where $EnvIn$ – the flux intensity of the $CO_2$ injection from the external environment, $EnvOut$ – the flux intensity of the $CO_2$ emission to the external environment, $\sum IntSrc_k$ – the total flux intensity of internal $CO_2$ sources, $\sum IntSink_l$ – the total flux intensity of internal $CO_2$ sinks.

The external components of CB and their effects are independent of the characteristics of the area of interest, unlike the internal components. The internal components of the CB clearly correspond to the components of the ecosystem - plants of certain species, soil, etc. This balance defines the total amount of $CO_2$ in the atmosphere of the area and consequently the *CDC=Func(CB)*.

The process of gas injection is inertial. For example, $CO_2$ emissions from a power plant do not change the CDC in every part

of the Earth's atmosphere, they only affect the neighbouring areas, and even then, it happens slowly, over some time. This

process is described by diffusion and environmental conditions. We assume that the CDC in a "small area" that was formed at some earlier time does not change significantly during the time it takes the satellite to measure the CDC in neighbouring "small areas", and interpret a data acquisition as a "monochrome image snapshot" of data.

The next two characteristics are also relevant to the definition of "small area". Firstly, the characteristics of physical and chemical inertness in the atmosphere and soils will lead to different spatial distributions of the characteristics, and the speed of these processes will affect the size of the cells by considering the value limit of the specified deviation. Secondly, in digital filtration, the size of the cells processed must be the same, which is limited by the size of the smallest area of the system. Another filtration requirement concerns the presence and location of neighbouring areas around the area of interest. According to the mathematical rules of sliding filtration (Aubry et al., 2014), cells should be located close to each other and partially have common boundaries, as shown in Fig. 1a. This requirement also leads to the neglect of air mass transport, as the short distances between the area of interest and neighbouring areas minimize its impact – transferred external air masses will give approximately the same $EnvIn$ and $EnvOut$ components in all neighbouring areas (in the filter focus). The small size and close location of cells also make it possible to detect the influence of external factors with synchronous changes in the monitored parameter with equal or proportional values (Fig. 1b).

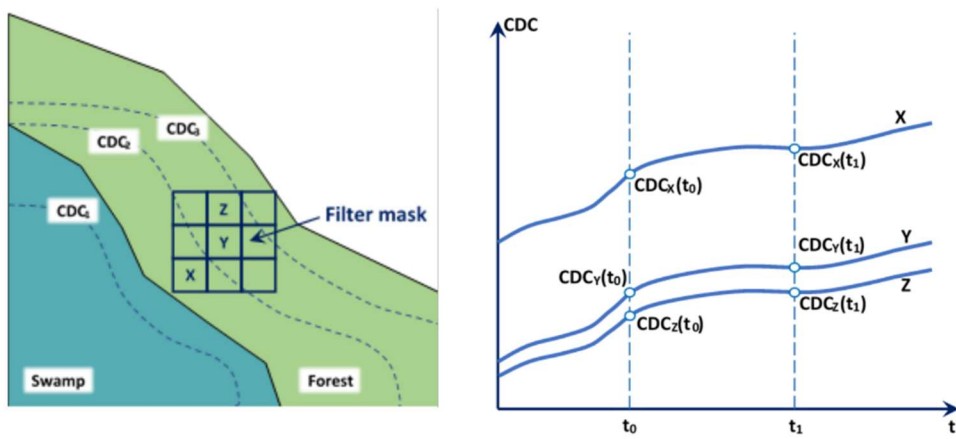

(a) An example of $CO_2$ source and sink location  (b) An example of synchronous CDC changes over time

**Figure 1: Spatial and temporal CDC changes.**

For example, at $t_0$, we expect different concentrations at points $X$, $Y$ and $Z$ – $CDC_X(t_0)$, $CDC_Y(t_0)$ and $CDC_Z(t_0)$, respectively, and assume that concentrations are related according to inequality (2):

$$CDC_X(t_1) > CDC_X(t_0), CDC_Y(t_1) > CDC_Y(t_0), CDC_Z(t_1) > CDC_Z(t_0) \tag{2}$$

Inequality (2) can be explained by natural processes ‑ continuous changes in temperature, humidity and other characteristics that lead to changes in the $CO_2$ emissions, e.g. from a swamp (Fig. 1a), and corresponding changes in the $CO_2$ concentration in neighbouring forest areas. Distance from the source or wind direction also affect the concentration. The time step for observing changes in $CO_2$ concentrations is 3 hours in the selected dataset.

If, at $t_1 > t_0$, the concentrations change according to (2) while all internal environmental conditions remain stable, this will result in a simultaneous multi-point ($X$-$Z$) increase in CDC as shown in (3):

$$\Delta CDC_X \simeq \Delta CDC_Y \simeq \Delta CDC_Z, \text{ where } \Delta CDC_{X,Y,Z} = CDC_{X,Y,Z}(t_1) - CDC_{X,Y,Z}(t_0) \tag{3}$$

Relationships 1-3 describe the connectivity and synchronicity of concentration change processes, but not their randomness. For example, Eq. (4) describes a synchronous increase in concentration due to daytime solar radiation, based on the conditions outlined in (3).

$$CDC_X(t_0) < CDC_X(t_1), CDC_Y(t_0) < CDC_Y(t_1), CDC_Z(t_0) < CDC_Z(t_1) \tag{4}$$

The above relationships and assumptions lead us to the conclusion that the $CO_2$ deltas shown in Eq. (4) correspond to the synchronous CDC changes for the whole area under the influence of external environmental conditions.

Based on Eq. (1), the difference between the CBs for two small neighbouring ecosystems can be described by Eq. (5):

$$CB_1 - CB_2 = (EnvIn_1 - EnvOut_1 + \sum IntSrc_{k1} - \sum IntSink_{l1}) - (EnvIn_2 - EnvOut_2 + \sum IntSrc_{k2} - \sum IntSink_{l2}) \tag{5}$$

According to the concept of small neighbouring areas, the values of *EnvIn* and *EnvOut* are equal in all cells, therefore the result of (5) can be interpreted as the difference in $CO_2$ fixation efficiency with (6):

$$CB_1 - CB_2 = (\sum IntSrc_{k1} - \sum IntSink_{l1}) - (\sum IntSrc_{k2} - \sum IntSink_{l2}) \tag{6}$$

If the characteristics of the neighbouring ecosystems are similar (each of the $IntSrc_{k1}$ sources of the first area is equal to $IntSrc_{k2}$ in the second neighbouring area, and each of the $IntSink_{l1}$ is equal to $IntSink_{l2}$), then based on (6) it is possible to identify the emergence of the external $CO_2$ source according to (7):

$$CB_1 - CB_2 = (EnvIn_1 - EnvOut_1) - (EnvIn_2 - EnvOut_2) \tag{7}$$

Each ecosystem is surrounded by neighbouring ecosystems, which can be represented in the Cartesian coordinate system with a set of indices in the vertical, horizontal and diagonal directions:

$$\begin{vmatrix} 4 & 3 & 2 \\ 5 & 0 & 1 \\ 6 & 7 & 8 \end{vmatrix} \tag{8}$$

When we form the convolutional filter of the difference between the central element and a given element, the coefficient "1" is placed in the centre of the matrix (zero index) and the coefficient "-1" is in the position defined by a given index. Using this indexing system and the convolutional filter principle, the difference (5) can be described in a matrix operation form over CB data as:

$$F(CB) = (CB_1 - CB_2) \Rightarrow (1 \times CB_1 + (-1) \times CB_2) \Rightarrow \begin{vmatrix} 0 & 0 & -1 \\ 0 & 0 & 1 \\ 0 & 0 & 0 \end{vmatrix} \tag{9}$$

The central index corresponds to the area of interest, and the rest are neighbouring areas. The matrix for evaluating the difference between all 8 neighbouring cells is as follows:

$$F(CB) = \sum_{i=1}^{i=8} (CB_0 - CB_i) \Rightarrow \nabla(CB) = \begin{vmatrix} -1 & -1 & -1 \\ -1 & 8 & -1 \\ -1 & -1 & -1 \end{vmatrix} \tag{10}$$

The area of interest is identified as a $CO_2$ sink or source based on its CDC in relation to that of the neighbouring areas. This means that the resolution of the dataset and the number of neighbouring areas define the area of identification. Depending on the expected sizes of $CO_2$ sinks and sources, the resolution of the dataset and the size of the matrix of coefficients can be adjusted. This option shows the universality of the proposed algorithm with respect to the sizes of $CO_2$ sources and sinks.

This matrix corresponds to the Laplacian convolutional filter. This is a second-order filter used for edge detection and feature extraction (Aubry et al., 2014). Unlike first-order filters, we do not need separate filters to detect and then combine vertical and horizontal edges, as the Laplacian filter detects all edges regardless of direction.

In order to apply the Laplacian filter to a CDC dataset formed by carbon balances, we performed a convolution operation, which mathematically means a combination of two matrices, in our case one containing the CDCs and the other – the filter coefficients. The convolution operation, represented by Eq. (11), involves sliding the filter over the dataset, multiplying the CDCs by the corresponding coefficients and adding them up. The result is a new dataset of the same size as the original, but the calculated CDC differences can be positive, negative or zero. A positive value after digital filtration means that the original CDC in the area of interest is greater than the average CDC in the neighbouring areas. This area is identified as containing the $CO_2$ source. Conversely, an area with a negative value is identified as containing a $CO_2$ sink. A zero value indicates $CO_2$ homogeneous areas.

$$CDC_{filtered} = \begin{bmatrix} CDC_4 & CDC_3 & CDC_2 \\ CDC_5 & CDC_0 & CDC_1 \\ CDC_6 & CDC_7 & CDC_8 \end{bmatrix} \times \begin{bmatrix} -1 & -1 & -1 \\ -1 & 8 & -1 \\ -1 & -1 & -1 \end{bmatrix} \qquad (11)$$

This filter, with a size of 3x3 cells, covers the area of 4.8x6.6 km when scanned with OCO satellites (OCO, 2015). It is optimal for our task in terms of processing time and computational complexity – 15 arithmetic operations for an area of interest, and does not require additional computational resources. This partially provides real-time computation for the 6 areas in the satellite scan area strip, which requires 90 operations per second.

The test results of the proposed algorithm (Appendix A) for $CO_2$ source and sink area detection show that it is sufficient for a rapid fire response or for a detailed subsequent study of the $CO_2$ fixation characteristics of the vegetation in the sink area. We do not consider $CO_2$ advection for the source area detection because the influence of air mass transport is small. It is close to 6% at a wind speed of 30 m/sec and a scanning time of 3 data rows by satellite for 1 second (OCO, 2015). This value is applicable for the tasks of rough $CO_2$ source and sink areas detection.

**Appendix A: Results of $CO_2$ source and sink areas detection with Laplacian filter**

According to the proposed method, the preliminary detection of $CO_2$ sources and sinks involves the following steps: 1. Digital filtration of the $CO_2$ concentrations in the area of interest and identification of the area as a source or sink by the sign ("+" is a source, "-" is a sink). 2. Comparison of the superimposed filtered $CO_2$ concentrations with fire fluxes in the area of interest. 3. Finding the area where two parameters are closely superimposed at their maximum intensities.

To test the proposed algorithm with a $CO_2$ source area detection, we chose a large fire event in the Serengeti National Park, Tanzania, which started on 22 July 2016 and lasted for 31 days. We used CDC values as an indicator of a fire area and CDC

spatial differences to detect area boundaries. For the experiment, we took the CDCs for 27 July 2016 (Weir and Ott, 2022), the fifth day after the fire had started, to avoid the influence of additional $CO_2$ from a previous fire event in the area. The CDC distribution for this date is shown in Fig. A1a, but it is not possible to see the clear boundaries of the area, because the spatial CDC differences are blurred. In order to detect the fire area boundaries, we applied the Laplacian filter, assuming that all the CDCs in the area were measured at the same time. The results are shown in Fig. A1b, where each cell has a different shading,

representing a change in CDC intensity. The dark shaded cells are defined as $CO_2$ sources.

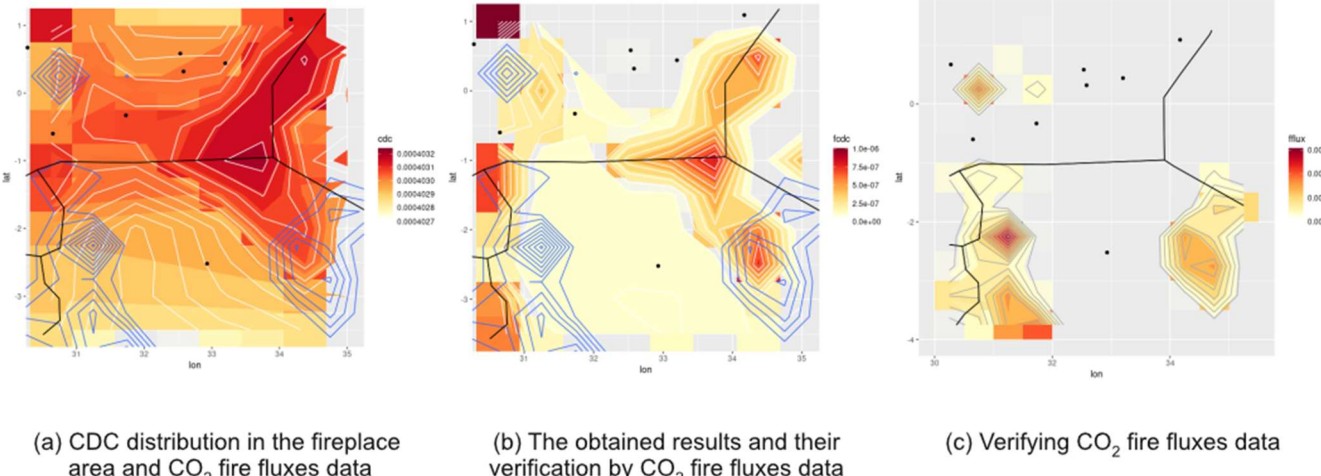

(a) CDC distribution in the fireplace area and $CO_2$ fire fluxes data

(b) The obtained results and their verification by $CO_2$ fire fluxes data

(c) Verifying $CO_2$ fire fluxes data

**Figure A1: Spatial distributions of the $CO_2$ parameters and the obtained results of the $CO_2$ source area detection.**

To verify the obtained results, we compared them with $CO_2$ fire fluxes data for the above-ground layer that contains daily fire emissions from 2003 to 2017 (Ott, 2020). These data are presented in Fig. A1c, which shows the $CO_2$ fire flux rate with colour

intensity and isolines, and in Figs. A1a, A1b in isolines only. The density of the isolines is related to the rate of flux intensity change – higher density corresponds to a higher rate, and lower density corresponds to a lower rate of change. The spatial resolution of the CDC dataset and the CDC fluxes dataset are different, 1°x1° and 0.5°x0.5° respectively. Different resolutions pose a challenge for source validation, so we use graphical image overlay with a relative placement by the object coordinates for a preliminary detection.

The greater the number of isolines around the point, the faster the concentration changed. The comparison of the experimental results and the flux data showed a rough agreement in the detection of the $CO_2$ source area. The differences in location can be explained by the higher spatial resolution of the flux data. However, the process of obtaining flux data requires either, a complex information model that is not real-time, or that a satellite to fly over the same point on Earth at least twice. In situations that require a more operational response, such as the start of a large forest fire near a populated area or an emergency at a

power plant with high $CO_2$ emissions, this may be too long. In our experiment, we chose available CDC data, interpreted as

"at the moment", and applied a Laplacian filter to detect $CO_2$ source areas. In reality, the proposed method can be applied to the satellite scanned data "strip" in real time.

Identifying areas that are $CO_2$ sinks is different from identifying areas that are short-term sources of $CO_2$. The most important terrestrial $CO_2$ sink is vegetation, the characteristics of which depend mainly on the time of day and the season. The size of large forests does not change over hours or days but over years or decades. We, therefore, need to define the boundaries of large forests once and then monitor them.

For our experiment, we chose CDC data (CAMS, 2020) for the period of active vegetation growth and analyzed data for Alaska in June 2016. We considered land cover (LC) type, biomass and growth phase (NDVI) as parameters of $CO_2$ fixation. First, we compared the CDC data processed with the Laplacian filter (Fig. A2a) with the LC types in Alaska. The results of this comparison are shown in Fig. A2b, where the LC data are presented in the FAO Land Cover Classification System (LCSS) (Friedl and Sulla-Menashe, 2019). The isolines in the figure show the change in CDC intensity, which roughly correspond to the formal boundaries of the LCCS vegetation classes. Forests with more than 60% tree cover (Di Gregorio, 2005): evergreen forests, deciduous forests and mixed forests show a higher $CO_2$ fixation.

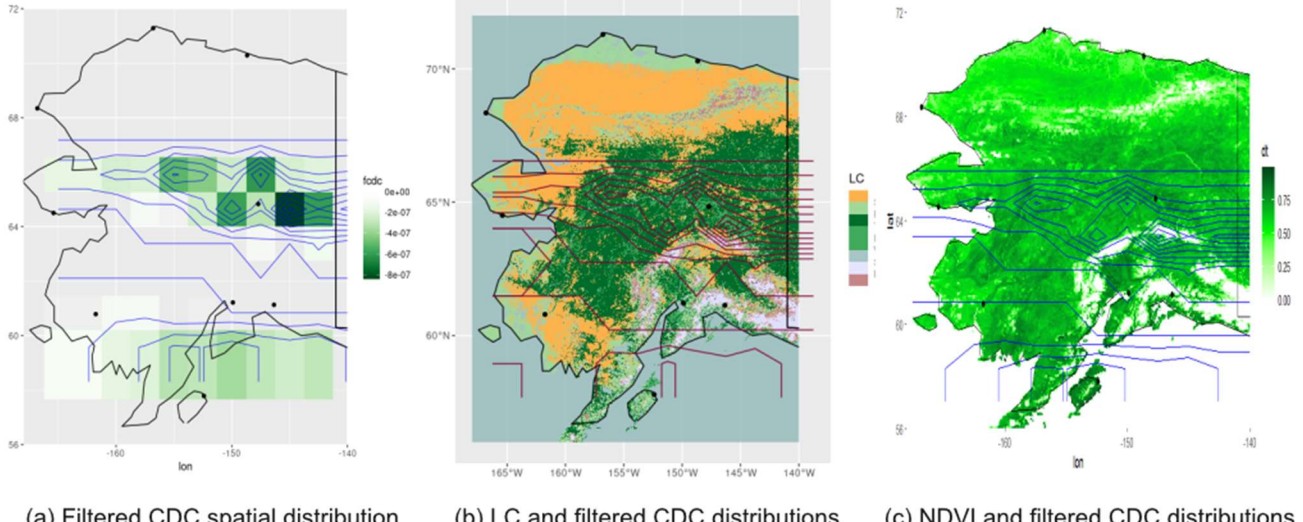

(a) Filtered CDC spatial distribution    (b) LC and filtered CDC distributions    (c) NDVI and filtered CDC distributions

**Figure A2: Spatial distributions of filtered CDC and vegetation indices for $CO_2$ sink area detection.**

In contrast, there is little spatial difference in $CO_2$ fixation between the areas covered by shrubs and herbs in Fig. A2b, possibly due to the small amount of biomass in these ecosystems and the potential influence of the nearby ocean.

The filtered CDC is nearly negligible on the mountain tops due to the uniform barren ground, ice, and snow zones. The NDVI (Fig. A2c) is also less significant in these areas. In contrast, the central part of Alaska, which is covered by a large amount of evergreen biomass with high NDVI (Didan, 2020) is identified as a $CO_2$ sink. Mountains protect this area from the influence of the oceans. These results could help in further work to explain the different $CO_2$ fixation potential in different subregions of these areas based on the absolute values of processed CDCs.

**Data availability**

Data are available at: https://doi.org/10.5281/zenodo.12532657.

**Author contributions**

YS conceptualised the study and designed the methodology; YS and VS processed the data; YS wrote the manuscript; VS and NW reviewed and partly edited the manuscript.

**Competing interests**

The contact author has declared that none of the authors has any competing interests.

**Acknowledgements**

This research was carried out in the SPACY group at the University of Tübingen (Germany), as part of "Research@Tübingen" fellowship and an individual project of the first co-author within the Philipp Schwartz scholarship program. Yana Savytska also acknowledges support from Prof. Dr. Kira Rehfeld, SPACY group leader. The authors also thank two anonymous reviewers and the editor, Min Chen, for the comments and editorial work, which helped to improve the manuscript.

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
