# Peer review of "ESD Ideas: Near real-time preliminary detection of carbon dioxide source and sink areas using a Laplacian filter"

_EGUsphere, 2024_

## Referee Comment (RC2)

In this work, the authors proposed a digital filtration method to detect CO2 source and sink regions. It's an interesting idea to introduce a tool used in Geosciences to the CO2 community. However, this new method described in the manuscript needs more details so the potential audience can fully comprehend.

(1) I find the method section is difficult to follow. I would suggest the authors to re-construct this section in a manner that: you first see a co2 concentration map (raw data); second, a pre-setup of the boundaries/masks, and basic information about your study region (topography, carbon ecosystem dynamics); third, how you apply your algorithm to detect sources and sinks by steps, and show intermediate figures to help the audience understand.

(2) Limitations of your methods need to be addressed. A big part of the work that I think is missing is the uncertainty and limitations of your method. For example, some regions are low-hanging fruit for detection, but some regions might be really difficult (topography, complex ecosystem). Also look at the CO2 concentration dataset, some regions show sufficient enhancement to be easily detected but some regions might look really uniform, how can these different scenarios be handled by your method? All of these need to be at least discussed in detail, so the readers can know how widely this method can be applied.

(3) Studies of CO2 point source quantification is well developed in the last couple years. This makes me wonder the meaning of only detecting sources or sinks but not quantifying them. Existing datasets like GPP or NPP are great proxies for this purpose and they serve many more scientific meaning. I would encourage the authors to describe in detail how this detection technique could uniquely provide more information, and if there is potential to further quantify the sources and sinks. I believe that would be a more appealing method to be used by our carbon community.

Minor comments:

Lines 11-12: No this is clearly not your research focus of this work.

Line 15: why a co2 concentration dataset is abbreviated as CDC?

Line 72: Eqs. (2) and (3) are identical.

---

## Author Comment (AC1)

**Reply to Referee 1**

We would like to thank Referee 1 for the constructive and positive view of our work. Below we address the comments on a point-by-point basis, aligning some of them with the comments of Referee 2.

**Response to comments in the summary**

**Referee:** In this work, the authors proposed a new method of using Laplacian filter on near real time $CO_2$ concentration data to qualitatively detect potential $CO_2$ sources and sinks in small areas.
This idea essentially boils down to using Laplacian filter to perform edge detection on digital images, and specifically in this work, the $CO_2$ concentration datasets are used as input digital images and the objects of interests are the $CO_2$ sources and sinks. The Laplacian filter are widely used in digital image processing/computer vision for edge detection purposes and generally performs well since the filter calculates the second derivatives of the given image and detects edges regardless of direction, but using the filter on $CO_2$ concentration data can impose some challenges including the shape of the object of interests can be often irregular and diffusive (as opposed to detecting made-made structures that often have crisp edges), and the spatial resolution of $CO_2$ concentration datasets.
Overall, it's good to introduce image-based methods of $CO_2$ sources and sinks detection to the general community of earth sciences, but revisions and clarifications are needed to resolve some confusions in the manuscript.

**Reply:** We agree with the comments on the specific challenges related to the characteristics and limitations of the method. However, we have stated in the paper that the $CO_2$ sinks and sources have been preliminary detected and that additional tools are needed to obtain more accurate results. The proposed digital filtration method is based on multiplication, difference and sum operations. All three operations can be applied to any $CO_2$ concentration value without mathematical limitations. A specific limitation of the digital filtration is described in lines 58-62. To avoid repetition, we will not include it in the *Anticipated changes*. Technical limitations in the resolution of satellite datasets (the resolution of a sensor) can pose an indirect challenge to preliminary detection, which can be partially overcome by matching the resolution of a dataset to the expected sizes of the areas to be detected.
Another challenge is the nature of the area of interest, which includes many individual characteristics, e.g. topography, complex ecosystem, natural or industrial origin, etc. These are more important in the following stages, which depend on the objectives and do not affect this preliminary stage. At the current stage of our work, we are exploring the ability of digital filters to capture and detect changes in various characteristics of natural processes, using $CO_2$ sinks and sources as an example, and focusing on the fact that these changes occur in near real-time and on their sign.

**Anticipated changes:** We will add the following sentences to the manuscript after line 18: "Applying digital filtration to $CO_2$ sinks and sources preliminary detection can be challenging due to their nature and behaviour. Industrial objects have more stable emission characteristics. Natural objects have a clear seasonal and also daily periodic dependence. This leads to the need for continuous observations in near real-time mode. Another potential challenge for satellite datasets are technical limitations in the resolution of satellite datasets (the resolution of a sensor), which indirectly challenge the preliminary detection. At the current stage of our work, we do not focus on the reasons that may affect the accuracy of detection, but aim to explore the ability of digital filters to capture and detect changes in various

characteristics of natural processes, for example, for the preliminary detection of $CO_2$ sinks and sources".

**Response to main comments:**

**Referee:** For the paragraph starting around line 70: Detailed assumptions are needed for equation (2): Why would you assume the inequality? As described in previous paragraphs, for the 'small area' and a short time period, if the emitting rate of $CO\_2$ is stable and the removal rate of $CO\_2$ is also stable (external and internal), why would $CDC(t\_1)$ be greater than $CDC(t\_0)$ at any given location (X, Y, or Z)?

**Reply:** A CDC measurement is performed for $\approx 0.3$ seconds (time for the satellite to scan all 9 areas in Fig. 1a), so the parameters from equation (1) can be considered constant during one measurement. The minimum time step between measurements in the dataset selected is 3 hours. Therefore, the concentrations at $t_0$ and $t_1$ in inequality (2) are different. Inequality (2) can be explained by natural processes - continuous changes in temperature, humidity and other characteristics leading to changes in the level of $CO_2$ emissions, e.g. from a swamp (Fig. 1a), and corresponding changes in $CO_2$ concentration in neighbouring forest areas. Distance from a source and wind direction also affect concentration.

**Anticipated changes:** We will change the paragraph before inequality (2) to: "For example, at $t_0$, we expect different concentrations at points X, Y and Z – $CDC_X(t_0)$, $CDC_Y(t_0)$ and $CDC_Z(t_0)$, respectively, and assume that concentrations are related according to inequality (2):".

And after inequality (2), we will add the following sentences: "Inequality (2) can be explained by natural processes – continuous changes in temperature, humidity and other characteristics that lead to changes in the $CO_2$ emissions, e.g. from a swamp (Fig. 1a), and corresponding changes in the $CO_2$ concentration in neighbouring forest areas. Distance from the source or wind direction also affect the concentration. The time step for observing changes in $CO_2$ concentrations is 3 hours in the selected dataset".

**Referee:** "Equation (1) and equation (2) seems identical, any reason why equation (2) needs to appear?"

**Reply:** We assume that the Referee had equations 2 and 3 in mind. Equation 2 is a mathematical interpretation of the dependence of the $CO_2$ data in Figure 1b. Equation 3 is an initial set of relationships that ground the relationships in Equation 4. So, they had different functional aims.

**Anticipated changes:** We will delete Equation 3, retain Equation 4 (#3 in the new numbering) and change the text of the explanatory paragraph after Equation 2 to: "If, at $t_1 > t_0$, the concentrations change according to (2) while all internal environmental conditions remain stable, this will result in a simultaneous multi-point (X-Z) increase in CDC as shown in (3)".

**Response to comments on the figures in the appendix:**

**Referee:** I am confused about the figures in appendix: how Figure A1 and Figure A2 are related? It seems Figure A2(a) is served as validation for results in Figure A1 (line 138) and Figure A2(b) and Figure A2(c) are for a completely different case study regarding $CO\_2$ sinks (line 158). If that's the case, why Figure A2(a) is together with and Figure A2(b) and Figure A2(c)?

**Reply:** We thank the Referee for pointing this out. Figures A1 and A2 are not related. Line 138 has a typo. It should read Figure A1(b) instead of Figure A2(a).

**Anticipated changes:** We will correct the numbering in Figure A1 along with the changes in the following comment.

**Referee:** Could you also clarify what are the isolines on both figures and the way to interpret? If the pixels in Figure A1(a) are already CO_2 concentrations, then how are the isolines calculated and what do those lines mean?

**Reply:** The blue (dark) isolines in Fig. A1(a) - Fig. 1(c) (in the new numbering) correspond to the intensities of the $CO_2$ fire fluxes plotted with the geom_contour function from the ggplot library in the R programming environment. The density of the isolines is related to the parameter change rate. Therefore, a higher density indicates a higher rate and a lower density indicates a lower rate of the $CO_2$ fluxes. The flux change rates are higher near the fireplaces, and the corresponding isolines are located closely. We use isolines to validate the fire source location by plotting filtered $CO_2$ concentrations and $CO_2$ fire fluxes in the same panel. Ideally, the highest density of the flux isolines should coincide with the darkest colour for the filtered $CO_2$ concentrations. The white (light) isolines in Fig. A1(a) and Fig. 1(b) are an additional interpretation of the $CO_2$ concentrations and filtered $CO_2$ concentrations, respectively, and help to see the changes in these parameters more precisely than with colour alone. The isolines in Figs. A2(a) - A2(c) (line 149) are also plotted with the geom_contour function and show the filtered $CO_2$ concentrations. These data are then superimposed with land cover (Fig. A2(b)) and NDVI (Fig. A2(c)) data for preliminary sink area detection.

**Anticipated changes:** For a better understanding of the algorithm, we will include another figure to show the distribution of the $CO_2$ fire fluxes with both colours and isolines in it. The proposed Fig. A1(c) is shown below together with the $CO_2$ distribution in the fire area in Fig. A1(a) and the obtained results in Fig. A1(b).

[Figure]

(a) CDC distribution in the
fireplace area and
CO2 fire fluxes data

(b) The obtained results and
their verification by
CO2 fire fluxes data

(c) Verifying CO2 fire fluxes data

Figure A1: Spatial distributions of the $CO_2$ parameters and the obtained results of the $CO_2$ source area detection

We will change the sentence "The flux data are presented in Fig. A2a with isolines showing the rate of CDC changes" in line 138 to: "These data are presented in Fig. A1c, which shows the $CO_2$ fire flux rate with colour intensity and isolines, and in Figs. A1a, A1b in isolines only. The density of the isolines is related to the rate of flux intensity change – higher density corresponds to higher rate and lower density corresponds to lower rate of change". We will also change the titles of Figure A2 (line 150). The new title of the general figure will be "Figure A2: Spatial distributions of vegetation indices and filtered CDC for $CO_2$ sink area detection". New title of Fig. A2(b) will be "Spatial distributions of LC and filtered CDC" and new title of Fig. A2(c) - "Spatial distributions of NDVI and filtered CDC".

**Response to specific notes:**

**Referee:** What is CDC? If it's CO_2 concentration dataset why it's not CCD?
**Reply:** The abbreviation CDC, used for the first time in line 15, stands for Carbon Dioxide Concentration.
**Anticipated changes:** none.
**Referee:** Line 138: For the CO_2 flux dataset (Lesley, 2020), that is spatial resolution of the datasets and how is it when compared with the CDC containing the fire event?
**Reply:** The spatial resolution of the CDC dataset and the CDC fire fluxes dataset are different - 1°x1° and 0.5°x0.5° respectively. The graphical image overlay with a relative placement by the object coordinates was used with ggplot's internal tools for a rough evaluation of the results.
**Anticipated changes:** We will add these explaining sentences after line 138: "The spatial resolution of the CDC dataset and the CDC fluxes dataset are different, 1°x1° and 0.5°x0.5° respectively. Different resolutions pose a challenge for source validation, so we use graphical image overlay with a relative placement by the object coordinates for a preliminary detection".

**Referee:** Does the flux dataset contains the fire event in 2016?
**Reply:** The flux dataset contains daily fire emissions over the period 2003 to 2017, including the selected fire event in 2016.
**Anticipated changes:** We will specify the details of the fire fluxes dataset in line 138: "To verify the obtained results, we compared them with a $CO_2$ fire fluxes data for the above-ground layer that contains daily fire emissions from 2003 to 2017 (Lesley, 2020)".

**Referee:** And how are the isolines calculated in the validation plot (Figure A2(a))?
**Reply:** In Fig. A2(a) (Fig. A1(b) in the new numbering), the isolines are plotted from the $CO_2$ fire flux dataset using the geom_contour function from the ggplot library in the R programming environment. The validation data of the fire placement are plotted with colour and isolines in the additional Fig. A1(c), with the numbering of the figures in Fig. A1 corrected.
The isolines in Figs. A2(a) - A2(c) (line 149) are also plotted with the geom_contour function, and show the filtered $CO_2$ concentrations. These data are then superimposed with land cover (Fig. A2(b)) and NDVI (Fig. A2(c)) data for preliminary sink area detection.
**Anticipated changes:** There are no additional changes as they are already included in the earlier comment on isolines.

---

## Author Comment (AC2)

**Reply to Referee 2**

We would like to thank Referee 2 for bringing important details of the proposed method to our attention. Below we address the comments, grouping them by their close relationship to each other and aligning some of them with the comments of Referee 1.

**Response to main comments**

**Referee:** I find the method section is difficult to follow. I would suggest the authors to re-construct this section in a manner that: you first see a co2 concentration map (raw data); second, a pre-setup of the boundaries/masks, and basic information about your study region (topography, carbon ecosystem dynamics); third, how you apply your algorithm to detect sources and sinks by steps, and show intermediate figures to help the audience understand.

**Referee:** Limitations of your methods need to be addressed. A big part of the work that I think is missing is the uncertainty and limitations of your method. For example, some regions are low-hanging fruit for detection, but some regions might be really difficult (topography, complex ecosystem). Also look at the CO2 concentration dataset, some regions show sufficient enhancement to be easily detected but some regions might look really uniform, how can these different scenarios be handled by your method? All of these need to be at least discussed in detail, so the readers can know how widely this method can be applied.

**Replies:** We agree with the Referee that the mathematical material describing the proposed method can be complex. However, according to the ESD Idea Paper format, authors should present a comprehensive analysis and justification in a few pages. We thank the Referee for the idea to restructure the paper, which can be applied in a full paper in the following stages of our research, and will add the algorithm for $CO_2$ sources and sinks detection to the *Anticipated changes* below.

We also agree with the comments on the specific challenges related to the characteristics and limitations of the method. However, we have stated in the paper that the $CO_2$ sinks and sources have been preliminary detected and that additional tools are needed to obtain more accurate results. The proposed digital filtration method is based on multiplication, difference and sum operations. All three operations can be applied to any $CO_2$ concentration value without mathematical limitations. A specific limitation of the digital filtration is described in lines 58-62. To avoid repetition, we will not include it in the *Anticipated changes*. Technical limitations in the resolution of satellite datasets (the resolution of a sensor) can pose an indirect challenge to preliminary detection, which can be partially overcome by matching the resolution of a dataset to the expected sizes of the areas to be detected.

Another challenge is the nature of the area of interest, which includes many individual characteristics, e.g. topography, complex ecosystem, natural or industrial origin, etc. These are more important in the following stages, which depend on the objectives and do not affect this preliminary stage. At the current stage of our work, we are exploring the ability of digital filters to capture and detect changes in various characteristics of natural processes, using $CO_2$ sinks and sources as an example, and focusing on the fact that these changes occur in near real-time and on their sign.

**Anticipated changes:** We will change the following sentence in the abstract (lines 6-8): "Here, we detect ecosystem areas with weighty changes in the $CO_2$ concentration using digital filtration, similar to image processing techniques, to identify terrestrial $CO_2$ sources and sinks" to: "Here, we propose digital filtration with a Laplacian filter for preliminary detection of areas with different changes in the characteristics of natural processes, using $CO_2$ sinks and sources as an example".

We will also add the following sentences to the manuscript after line 18: "Applying digital filtration to $CO_2$ sinks and sources preliminary detection can be challenging due to their nature and behaviour. Industrial objects have more stable emission characteristics. Natural objects have a clear seasonal and also daily periodic dependence. This leads to the need for continuous observations in near real-time mode. Another potential challenge for satellite datasets are technical limitations in the resolution of satellite datasets (the resolution of a sensor), which indirectly challenge the preliminary detection. At the current stage of our work, we do not focus on the reasons that may affect the accuracy of detection, but aim to explore the ability of digital filters to capture and detect changes in various characteristics of natural processes, for example, for the preliminary detection of $CO_2$ sinks and sources".

We will add the following detection algorithm at the beginning of the Appendix: "According to the proposed method, the preliminary detection of $CO_2$ sources and sinks involves the following steps: 1. Digital filtration of the $CO_2$ concentrations in the area of interest and identification of the area as a source or sink by the sign ("+" is a source, "-" is a sink). 2. Comparison of the superimposed filtered $CO_2$ concentrations with fire fluxes in the area of interest. 3. Finding the area where two parameters are closely superimposed at their maximum intensities".

For a better understanding of the algorithm, we propose to include another figure showing the distribution of the $CO_2$ fire fluxes with both colours and isolines. The proposed Fig. A1(c) is shown below, together with the distributions of the $CO_2$ parameters in the fire area in Fig. A1(a) and the obtained results in Fig. A1(b).

[Figure]

(a) CDC distribution in the fireplace area and CO2 fire fluxes data

(b) The obtained results and their verification by CO2 fire fluxes data

(c) Verifying CO2 fire fluxes data

Figure A1: Spatial distributions of the $CO_2$ parameters and the obtained results of the $CO_2$ source area detection

We will also change the sentence "The flux data are presented in Fig. A2a with isolines showing the rate of CDC changes" in line 138 to: "These data are presented in Fig. A1c, which shows the $CO_2$ flux rate with colour intensity and isolines, and in Figs. 1a, 1b with isolines only. The density of the isolines is related to the rate of flux intensity change – higher density corresponds to higher rate, and lower density corresponds to lower rate of change".

**Referee:** Studies of CO2 point source quantification is well developed in the last couple years. This makes me wonder the meaning of only detecting sources or sinks but not quantifying them. Existing

datasets like GPP or NPP are great proxies for this purpose and they serve many more scientific meaning. I would encourage the authors to describe in detail how this detection technique could uniquely provide more information, and if there is potential to further quantify the sources and sinks. I believe that would be a more appealing method to be used by our carbon community.

**Reply:** Functional indices such as GPP or NPP can be good proxies for detecting long-term continuous sources and sinks of $CO_2$, assessing long-term $CO_2$ processes, and quantifying the effects of changes in $CO_2$ after these changes have occurred. These and similar indices are more integral and reflect the cumulative result over a period of time with large measurement inertia. For example, it is rather difficult to use them to determine how $CO_2$ fixation changes during the day (this is one of the tasks of our main project), which, in our opinion, is easier to do by analysing fluctuations in atmospheric $CO_2$. Also, the detection of $CO_2$ sources and sinks can be more meaningful than their quantification for near real-time decision making based on $CO_2$ dynamics. For example, detecting a short-term $CO_2$ source in the forest area without quantification may help stop a forest fire in its early stages. Near real-time detection of a $CO_2$ sink area can help understand the time frames of different sink activity periods and capture transitions between periods. These are potential applications of the proposed method.

**Anticipated changes:** We will add the following sentences to line 29: "In our paper, we do not quantify $CO_2$ sources and sinks, because quantification is valuable for understanding the consequences of $CO_2$ changes after these changes have occurred. Our focus is on short-term (e.g., hours) $CO_2$ changes, which can help detect $CO_2$ sources and sinks and their different phases of development in near real-time, until further analyses can be performed".

**Response to minor comments**

**Referee:** Lines 11-12: No this is clearly not your research focus of this work.

**Reply:** In this idea paper, we propose and justify the applicability of the algorithm for the preliminary detection of $CO_2$ sources and sinks, which is one of the tasks of the new method. Another feature of this algorithm is the possibility of using it in near real-time.

**Anticipated changes:** We will specify that the proposed detection algorithm is part of a new method for $CO_2$ reduction and make the following changes to lines 11-13: "Our primary research therefore focuses on the development of a new method for $CO_2$ reduction. As part of this method, we propose an algorithm for the near real-time preliminary detection of $CO_2$ source and sink areas. This algorithm can help to facilitate the monitoring, reporting and verification of $CO_2$ source and sink areas".

**Referee:** Line 15: why a co2 concentration dataset is abbreviated as CDC?

**Reply:** The abbreviation CDC stands for Carbon Dioxide Concentration, used first in the line 15.

**Anticipated changes:** none.

**Referee:** Line 72: Eqs. (2) and (3) are identical.

**Reply:** Equation 2 is a mathematical interpretation of the dependence of the $CO_2$ data on Figure 1b. Equation 3 is an initial set of relationships that ground the relationships in Equation 4. They therefore had different functional aims.

**Anticipated changes:** We will delete Equation 3, retain Equation 4 (#3 in the new numbering) and change the text of the explanatory paragraph after Equation 2 to: "If, at $t_1 > t_0$, the concentrations change according to (2) while all internal environmental conditions remain stable, this will result in a simultaneous multi-point (X-Z) increase in CDC as shown in (3)".